



**Indirect assimilation of radar reflectivity data with an adaptive hydrometer retrieval scheme for the short-term severe weather forecasts**

Lixin Song [1,2,3], Feifei Shen[1,2,4,5*], Zhixin He[6], Dongmei Xu[1], Aiqing Shu[1], and Jiajun Chen[1]

[1] Key Laboratory of Meteorological Disaster, Ministry of Education (KLME) /Joint International Research Laboratory of Climate and Environment Change (ILCEC) /Collaborative Innovation Center on Forecast and Evaluation of Meteorological Disasters (CIC-FEMD), Nanjing University of Information Science & Technology, Nanjing 210044, China

[2] China Meteorological Administration Tornado Key Laboratory

[3] Department of Atmospheric and Oceanic Sciences and Institute of Atmospheric Sciences, Fudan University, Shanghai 200433, China

[4] China Meteorological Administration Radar Meteorology Key Laboratory, Nanjing 210000, China

[5] Shanghai Typhoon Institute, China Meteorological Administration, Shanghai 200030, China

[6] Anhui Meteorological Observatory, Hefei 230000, China

*Corresponding author address:
Feifei Shen
Nanjing University of Information Science & Technology
ffshen@nuist.edu.cn





## Abstract


Different hydrometeor retrieval schemes are explored based on the Weather Research and
Forecasting (WRF) model in the indirect assimilation of radar reflectivity for two real cases
occurred during June 2020 and August 2018. When retrieving hydrometeors from radar reflectivity,
there are two commonly used hydrometeor classification methods: "temperature-based" and
"background hydrometer-dependent" schemes. The hydrometeor proportions are usually
empirically assigned in the "temperature-based" method within different background temperature
intervals. Whereas, in the "background hydrometer-dependent" scheme, each type of the
hydrometeor is derived based on the portions estimated from the background field for different radar
reflectivity ranges. In this study, a blending scheme is designed to combine "temperature-based"
and "background hydrometer-dependent" methods adaptively to avoid errors caused by fixed
relationships and reduce uncertainties introduced by the background field itself. Three experiments,
EXP_temp, EXP_bg, and EXP_temp-bg are conducted using the "temperature-based" method,
"background hydrometer-dependent" scheme, and blending scheme, respectively. It is found that,
the blending scheme facilitates the generation of accurate hydrometeor species which will enhance
the effectiveness of radar data assimilation. EXP_temp-bg is capable of analyzing radar reflectivity
structures more accurately compared to both EXP_temp and EXP_bg. Besides, due to the adaptive
combination of "temperature-based" and "background hydrometer-dependent" schemes, the
EXP_temp-bg experiment predict the radar reflectivity structures and precipitation intensity more
accurately.
**Key words**: Numerical weather prediction, Radar data assimilation, Hydrometeor retrieval.

## 1. Introduction

The initial condition is a crucial factor in enhancing the accuracy of numerical weather prediction
(Navon, 2009; Kain et al., 2010; Lopez, 2011; Xu et al., 2021). Compared to conventional
observations, doppler radar observations have extremely high temporal and spatial resolution, as
well as containing precipitating hydrometeor information (Zhao et al., 2012; Li et al., 2013; Kong
et al., 2020). Therefore, radar is one of the key platforms for obtaining proper initial conditions to
successfully predict convective storms (Lilly, 1990; Dawson et al., 2015; Gustafsson et al., 2018;



Shen et al., 2020a; Xu et al., 2022; Chen et al., 2023). A number of efforts have been devoted to
assimilating radar data into mesoscale numerical models (Lindskog et al., 2004; Dowell et al., 2011;
Sun et al., 2014; Bick et al., 2016; Tong et al., 2020; Shen et al., 2016, 2019, 2022; Wan et al., 2024).
Radar observations have two fundamental variables: radar radial velocity (Vr) and radar
reflectivity (Z). Assimilating radar radial velocity is conducive to improving the dynamical structure
of the initial field. Numerous scholars are dedicated to researching radar radial velocity assimilation
(Gao et al., 2004; Simonin et al., 2014; Li et al., 2016; Shen et al., 2020b). Based on the three-
dimensional variational (3DVar) system of the fifth generation Pennsylvania State University-
NCAR Mesoscale Model (MM5), Xiao et al. (2005) developed a radar radial velocity observation
operator, and investigated the impact of assimilating radar radial velocity on precipitation forecasts.
Besides, Wang et al. (2013b) employed the four-dimensional variational (4DVar) system to
assimilate radar radial velocity and reflectivity into the model for enhancing forecasting accuracy.
In contrast, assimilating radar reflectivity is more challenging than assimilating the radial wind,
on account of its highly nonlinear observation operator and close relationship with complex
microphysics (Borderies et al., 2019; Xu et al., 2019). Currently, there are two main methods for
assimilating radar reflectivity: direct assimilation and indirect assimilation. Xiao et al. (2007)
proposed a direct assimilation scheme for radar reflectivity based on the 3DVar system of MM5.
The water content was classified according to phases using warm rain microphysical processes.
However, due to the absence of ice phase particles, the positive impact is not promising in cases of
deep moist convections generated through cold-cloud processes. To assimilate radar reflectivity into
numerical weather prediction (NWP) models more effectively, Gao and Stensrud (2012) proposed
a hydrometeor classification method based on the 3DVar system in the direct assimilation of radar
reflectivity. The results demonstrated that this classification method benefits to accelerate the
convergence speed of the analysis and reduce errors in the analysis. Compared to variational data
assimilation methods, Ensemble Kalman Filter (EnKF; Evensen, 1994) is a better choice for
assimilating radar reflectivity directly, since EnKF does not require consideration of the tangent or
adjoint model of the observation operator (Liu et al., 2019). Based on the EnKF method, Tong and
Xue (2005) assimilated the simulated radar observations from a supercell storm. The results
indicated that directly assimilating radar reflectivity data has a positive impact on both analyses and





forecasts. Although the forward operator of reflectivity tends to be easily implemented in EnKF, its
computational cost is too high to be widely applied in the scientific research and operational
forecasting (Kong et al., 2018).

To avoid the issue of high nonlinearity in radar reflectivity observation operators, the indirect

assimilation method is often used in the NWP. Based on the Advance Regional Prediction System
(ARPS), Hu et al. (2006) investigated the impact of cloud analysis using radar reflectivity on
forecasting tornado storms. They found that cloud analysis helps to adjust the temperature, humidity
fields, and hydrometeors within the clouds, thereby improving tornado predictions. Also,
Schenkman et al. (2011) found that cloud analysis technology is able to adjust cloud variables to
better suit the dynamic and thermal fields. However, cloud analysis schemes rely largely on
uncertain empirical relationships, thus usually hardly suppressing the generation of spurious echoes.
Using the 4DVar system, Sun and Crook (1997) proposed to assimilate rainwater mixing ratios
retrieved from reflectivity instead of directly assimilating reflectivity, which seems to produce better
analysis results. Based on the 3DVar system of WRF, Wang et al. (2013a) further demonstrated that
assimilation of rainwater and estimated water vapor obtained from radar reflectivity reduces the
linearization error of the radar reflectivity observation operator, thus improving precipitation
forecasts. However, both indirect assimilation methods under the two variational frameworks are
employed in the warm-rain scheme, which restricts their applications above troposphere or in the
coexistence of liquid and ice particles. Shen et al. (2021) added hydrometeor control variables
included ice-phase particles when indirectly assimilating radar reflectivity observations of
Hurricane IKE, which enables track and intensity forecasts of the hurricane to be greatly improved.
For the indirect assimilation of radar reflectivity, one of the challenges is how to correctly classify
hydrometeors in observations. There are currently two methods to distinguish hydrometeor types.
One is to classify hydrometeor types according to background temperature (hereafter called
temperature-based) developed by Gao and Stensrud (2012), with fixed parameters and empirical
relations. Another is the "background hydrometer-dependent" hydrometeor retrieval scheme (Chen
et al., 2020, 2021). The "background hydrometer-dependent" method calculates hydrometeor
weights in various thresholds from the model background field to better allocate radar reflectivity
observation information. This approach avoids empirical thresholds and weighting coefficients



given in the "temperature-based" method, and benefits to improve the accuracy of hydrometeor
retrievals. However, the "background hydrometer-dependent" scheme also relies on the accuracy of
the background field itself. When the background field is similar to the observation, the "background
hydrometer-dependent" method tends to provide accurate hydrometeor weights. On the other hand,
when the background field differs significantly from the observation, the algorithm may not be
suitable for appropriately allocating hydrometeors of the radar reflectivity observation. Considering
their own limitations in either "temperature-based" or "background hydrometer-dependent"
schemes, this study aims to adaptively combine two above methods of classifying hydrometeors to
assimilate radar reflectivity more reasonably.
In the study, the WRF-3DVar methods, observation operators, and different retrieval methods are
included in the section 2. The section 3 shows experimental designs. The section 4 presents analysis
and forecast results of all experiments. The conclusion is presented in the section 5.

## 2. Methods

### 2.1 The WRF-3DVar system

Based on the incremental method proposed by Courtier et al. (1994), 3DVar uses the minimization
algorithm to solve the objective function. The cost function is as follows:

$$J = \frac{1}{2}(x - x_b)^T B^{-1}(x - x_b) + \frac{1}{2}[y_o - H(x)]^T R^{-1}[y_o - H(x)]. \tag{1}$$

The vectors $x$, $x_b$, and $y_o$ stand for analysis variables, background variables, and observation
variables. $B$ is the background error covariance, which is calculated by the National
Meteorological Center (NMC; Parrish and Derber, 1992) method. $R$ represents the observation
error covariance. $H$ is the nonlinear observation operator.

### 2.2 The radical velocity observation operator

The radial velocity observation operator is as follows:

$$V_r = u\frac{x - x_i}{r_i} + v\frac{y - y_i}{r_i} + (w - v_T)\frac{z - z_i}{r_i}. \tag{2}$$

$u$, v, and $w$ denote the zonal, meridional, and vertical wind component, respectively. $(x, y, z)$
and $(x_i, y_i, z_i)$ represent the radar position and observation position, respectively. $r_i$ is the
distance between the radar and the observation. $v_T$ is the terminal speed.



## 2.3 The radar reflectivity observation operator

According to Tong and Xue (2005), the radar reflectivity observation operator is as follows:

$$Z = 10 * log_{10}(Z_e), \tag{3}$$

$$Z_e = Z_e(q_r) + Z_e(q_s) + Z_e(q_g), \tag{4}$$

$$Z_e(q_x) = a_x(\rho q_x)^{1.75}. \tag{5}$$

$q_x$ means hydrometeor mixing ratios. $Z_e(q_x)$ (units: dBZ) is the equivalent reflectivity factor of rainwater, snow, and graupel. $a_x$ represents the fixed coefficient that is determined by the dielectric coefficient, density and intercept parameter of each hydrometeor. $\alpha_r$ is $3.63\times10^9$. For snow and graupel, the coefficient is temperature dependent. When the environmental temperature is greater than 0°C, $\alpha_s$ for wet snow is $4.26\times10^{11}$ and $\alpha_g$ for wet graupel is $9.08\times10^9$. When the temperature is below 0°C, $\alpha_s$ for dry snow is $9.80\times10^8$ and $\alpha_g$ for dry graupel is $1.09\times10^9$. $\rho$ is the air density. During the direct assimilation of radar reflectivity, the linearization errors are almost inevitable.

The indirect method assimilates the retrieved hydrometeors from the radar reflectivity. Firstly, it is required to determine the proportion of each hydrometeor in radar reflectivity observation. At present, there are two methods to obtain the proportion of each hydrometeor.

### 2.3.1 The "Temperature-based" method

In Gao and Stensrud (2012), the hydrometeor types in reflectivity are classified based on the background temperature. The specific values are as follows:

$$C_r = 1, C_s = C_g = 0, T_b > 5℃, \tag{6}$$

$$C_r = \frac{T_b+5}{10}, C_s = (1 - C_r) \cdot \frac{\alpha_s}{\alpha_s+\alpha_g}, C_g = (1 - C_r) \cdot \frac{\alpha_g}{\alpha_s+\alpha_g}, -5℃ < T_b < 5℃, \tag{7}$$

$$C_r = 0, C_s = \frac{\alpha_s}{\alpha_s+\alpha_g}, C_g = \frac{\alpha_g}{\alpha_s+\alpha_g}, T_b < -5℃. \tag{8}$$

$C_r$, $C_s$, and $C_g$ denote the weights of rainwater, snow, and graupel, respectively. $\alpha_r$, $\alpha_s$, and $\alpha_g$ represent the fixed coefficients of rainwater, snow, and graupel, respectively (Same as above). $T_b$ is the background temperature.

### 2.3.2 The "Background hydrometer-dependent" method

It is found that hydrometeor weights derived from the background field vary with individual


weather conditions, which helps to reduce errors resulting from fixed coefficients in Chen et al.
(2020, 2021). The specific process of calculating proportions is as follows:
(1) Compute the average equivalent radar reflectivity of each hydrometeor in different reflectivity
ranges and model layers based on the background field statistics.
(2) Calculate the weight of each hydrometeor in the background field.
(3) Divide radar reflectivity observations based on the weights derived from Step 2. If the
background field has missing data, the calculated climatological mean for one month will be
used instead.
### 2.3.3 The blending method
The blending method aims to utilize the two methods of partitioning hydrometeors accordingly
to retrieve muti-hydrometer more reasonably in radar reflectivity indirect assimilation. The formulas
are as follow:
$$\beta = \frac{\delta_t^2}{\delta_t^2 + \delta_b^2}, \tag{9}$$

$$C_x = \beta C_x^b + (1 - \beta) C_x^t. \tag{10}$$

$\delta_t^2$ represents the deviation between the hydrometeor content of the background field and the
retrieved hydrometeor content based on the "temperature-based" scheme. $\delta_b^2$ is the deviation
between the hydrometeor content of the background field and the retrieved hydrometeor by the
"background hydrometer-dependent" scheme. $C_x^t$ and $C_x^b$ are the weights calculated by the
"temperature-based" and "background hydrometer-dependent" methods, respectively.
# 3. Experimental design
WRF v4.3 and its data assimilation system WRFDA v4.3 are used in this study. Two convective
cases are studied in the study: 14 June in 2020 (called Case 1; Fig. 1a) and 6 August in 2018 (denoted
as Case 2; Fig. 1b). The specific applications of physical parametrizations are as follows: the WRF
Double-Moment 6-Class Microphysics (WDM6) scheme, the Rapid Radiative Transfer Model
(RRTM) long wave radiation scheme (Mlawer et al., 1997), the Dudhia short-wave radiation scheme
(Dudhia, 1989), the Yonsei University (YSU) boundary layer scheme (Hong et al., 2006), and the
Noah Land Surface Model (Chen and Dudhia, 2001) for land surface process scheme. No cumulus
parameterization scheme is employed. As shown in Table 1, three data assimilation (DA)
experiments are conducted to evaluate the effects of all retrieval methods in the study. For all three




DA experiments, the initial and lateral boundary conditions are provided by the NCEP Global
Forecast System (GFS) data. Besides, the specific flowchart is presented in the Fig. 2.



Table 1. The list of DA experiments.

| Experiments | Hydrometeor retrieval methods |
|---|---|
| EXP_temp | The "temperature-based" method |
| EXP_bg | The "background hydrometer-dependent" method |
| EXP_temp-bg | The blending method |

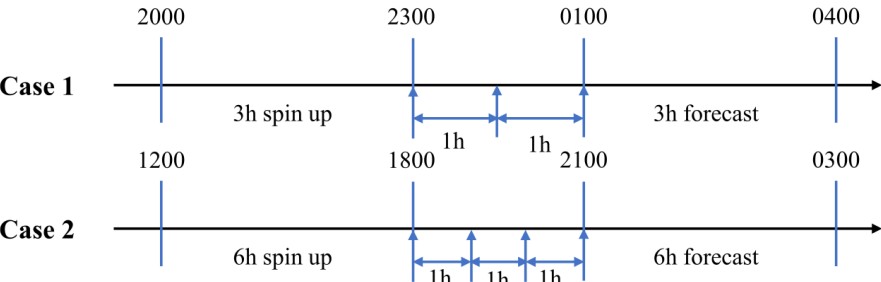


Fig. 1. The simulated area of (a) Case 1 and (b) Case 2, with the detecting ranges of the Nanjing radar and
Shenyang Radar.

Fig. 2. The assimilation flow charts of Case 1 and Case 2.



## 4. Experimental results

### 4.1 14 June 2020 case

Fig. 3 shows the observed reflectivity at 2300 UTC on 14 June, 0000 UTC, and 0100 UTC on 15 June 2020. At the beginning, there are strong echoes in the southwestern boundary of Jiangsu Province. Subsequently, the strong convective band begins to expand in both eastward and westward directions, stretching to the central Anhui Province and Jiangsu Province.

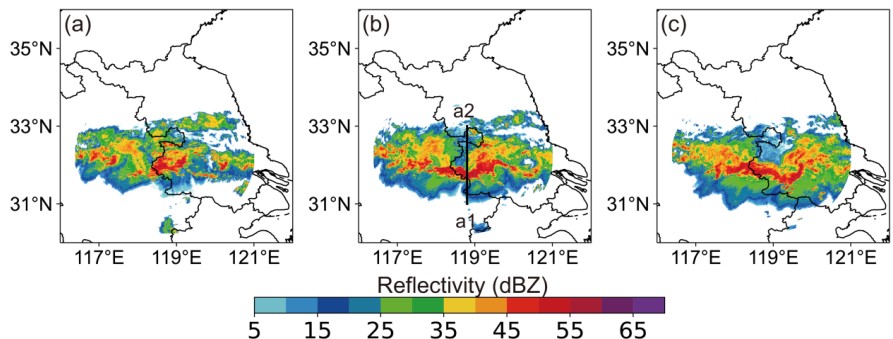

Fig. 3. The observed composite reflectivity fields (units: dBZ) at (a) 2300 UTC 14 June, (b) 0000 UTC, and (c)

0100 UTC 15 June 2020. The black line a1-a2 in the Fig. 3b is the vertical cross section location of Fig. 4.

Fig. 4 compares the Hydrometeor Classification Algorithm (HCA) based on dual-polarization radar observations with the hydrometeor retrieval results from the three experiments at 1500 UTC on June 14, 2020. The HCA diagram indicates that rainwater dominates the lower levels, while dry snow and graupel prevail at higher levels, with wet snow present near the melting layer. In the vertical cross sections of the three experiments (Figs. 4b, c, d), the overall distribution patterns of the retrieved hydrometeors appear reasonable, especially for rain and snow. Notably, the wet snow and graupel retrieved by EXP_temp-bg are more consistent with the HCA results compared to EXP_temp and EXP_bg.


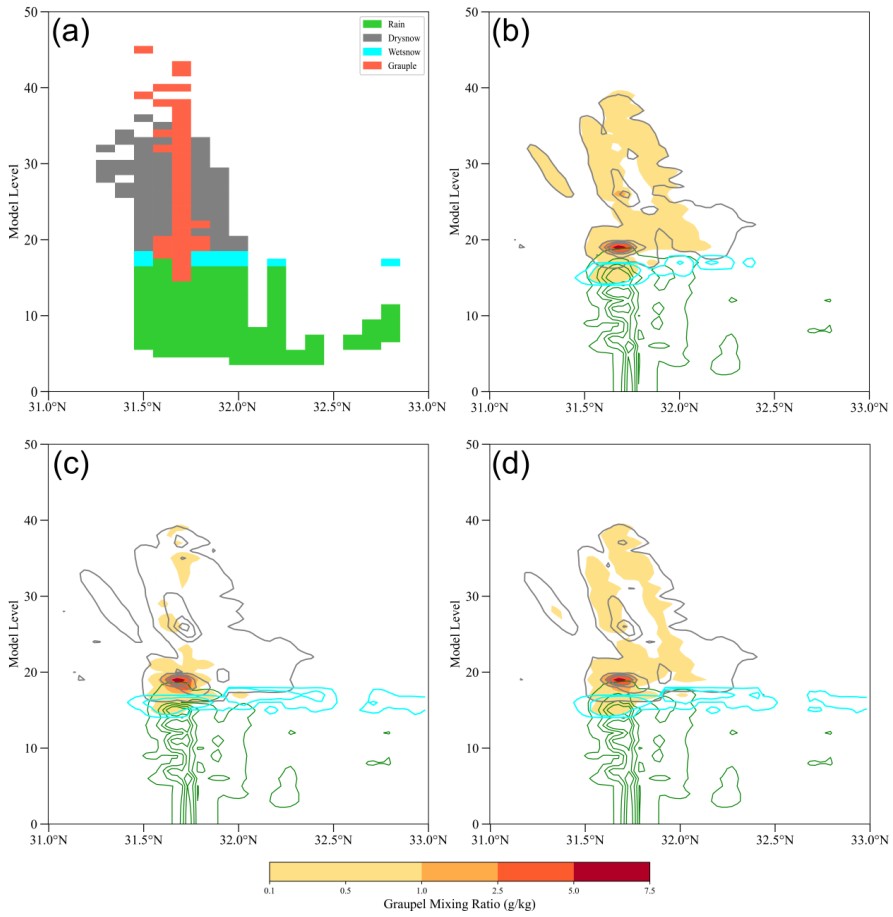

Fig. 4. The vertical sections of (a) hydrometeor classification algorithm based on the dual-polarization radar observations and retrieved hydrometeors for (b) EXP_temp, (c) EXP_bg and (d) EXP_temp-bg along the black lines a1-a2 at 1500 UTC. The retrieved hydrometeors refer to rainwater mixing ratio (green contours; units: dBZ), dry snow mixing ratio (grey contours; units: dBZ), wet snow mixing ratio (cyan contours; units: dBZ), and graupel mixing ratio (shading; units: dBZ), respectively.

To investigate the impact of the radar reflectivity DA based on the three hydrometeor retrieval methods, Fig. 5 shows the predicted composite reflectivity initiated at 0100 UTC 15 June. It is shown that the convective structure is divided into two parts (labeled C and D). From the observations (Fig. 3a), the combination of C and D is initially located in the western Jiangsu and eastern Anhui. Soon after, region D gradually separates from C and shifts eastward, displaying the reduced intensity and poor organization. At 0115 UTC, all DA experiments are able to capture region

C and region D, albeit with slightly weaker intensity compared to the observations. At 0130 UTC,
the patterns of region C predicted by three experiments are depart from the observation, while the
echoes for EXP_temp-bg exhibit the best organization. At 0145 UTC, the regions C in EXP_temp
and EXP_bg show a poor agreement with the observations. In contrast, EXP_temp-bg provides
more accurate forecast in terms of shape and intensity. At 0200 UTC, three experiments can predict
region C and region D to some extent, but region D in EXP_temp-bg has most accurate echo pattern.
In general, the blending scheme is conducive to improving the radar reflectivity forecast skill.

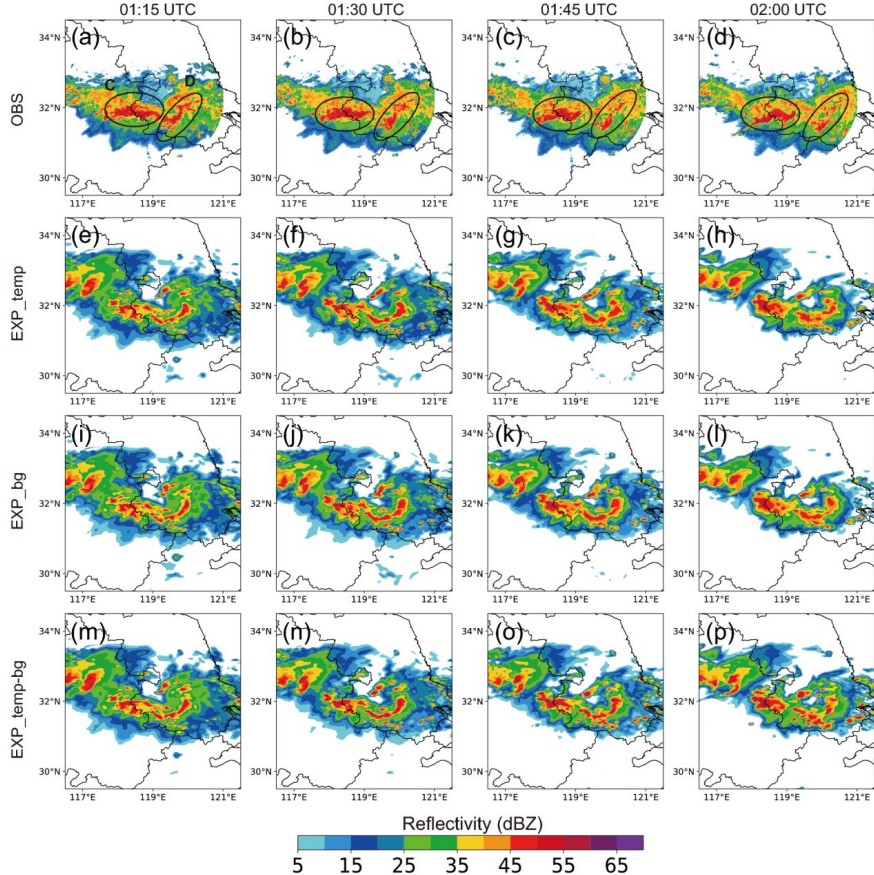


Fig. 5. The composite reflectivity (shaded; units: dBZ) predicted by (e)-(h) EXP_temp (i)-(l) EXP_bg and (m)-(p)
EXP_temp-bg for the 1-h forecast beginning at 0100 UTC 15 June 2020, as compared to (a)-(d) the observed
composite reflectivity. The labels C and D present the convection locations.
Fig. 6 displays the vertical cross sections of the relative humidity, radar reflectivity, and wind
fields at 1501 UTC. After 1-hour forecast, the cross sections from all experiments indicate the
presence of saturated water vapor columns near the strong echoes (around 32°N). Notably,
EXP_temp-bg also reveals a robust updraft, facilitating the transport of water vapor from lower to
upper levels. In comparison, EXP_temp-bg producess the most consistent thermal and dynamical
conditions, resulting in most accurate forecast of the convection.

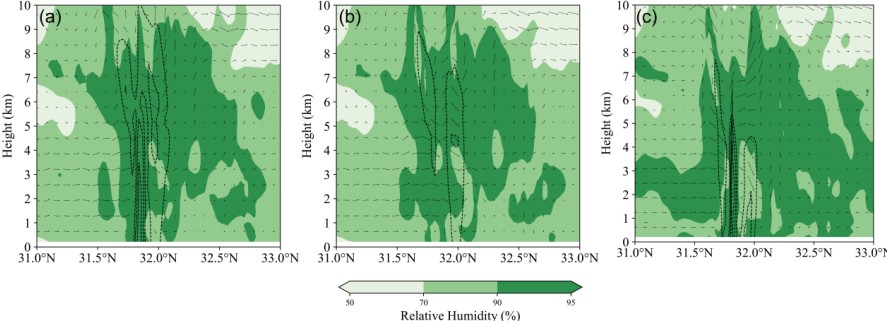


Fig. 6. The cross sections of relative humidity (shading; units: %), radar reflectivity (black contours starting at 40
dBZ; units: dBZ), and wind vectors for (a) EXP_temp, (b) EXP_bg and (c) EXP_temp-bg along the line a1-a2.
These are 1-hour forecasts initialized at 1501 UTC.
Fig. 7 shows the 3-h accumulated precipitation forecast from 1501 UTC to 1504 UTC on 15 June
2020. As depicted in Fig. 7a, the primary precipitation zone is concentrated along the western
boundary of Jiangsu Province, with accumulated precipitation exceeding 50mm. The precipitation
intensity is overestimated for three DA experiments. However, EXP_temp-bg effectively suppresses
two false precipitation areas, leading to the improved precipitation forecast.

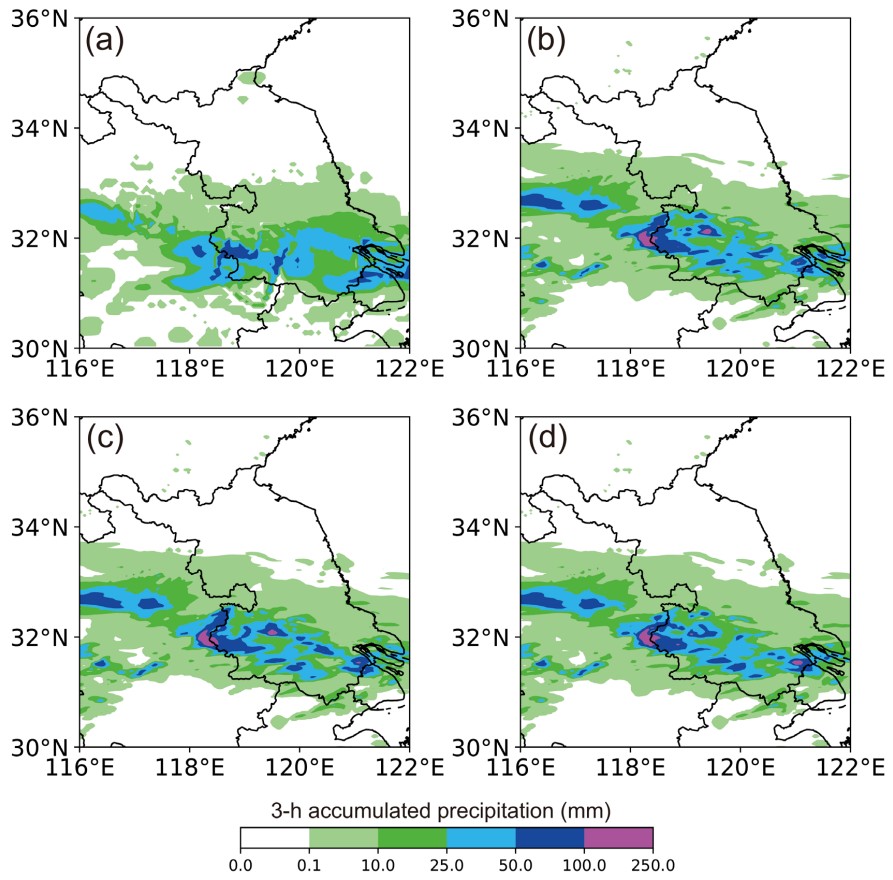

3-h accumulated precipitation (mm)

0.0    0.1    10.0    25.0    50.0    100.0    250.0

Fig. 7. 3-h accumulated precipitation valid at 0100 UTC 15 June 2020. (a) the observation, (b) EXP_temp, (c)

EXP_bg, and (d) EXP_temp-bg.

To quantitatively assess the performance of different hydrometeor retrieval schemes, the equitable

threat scores (ETS) are calculated for 0-3 h precipitation forecasts in EXP_temp, EXP_bg, and

EXP_temp-bg (Fig. 8). It is evident that as the precipitation threshold increases, the ETS values for

all three experiments decline progressively. Furthermore, EXP_temp and EXP_bg exhibit

comparable ETS values under various precipitation thresholds. In contrast, EXP_temp-bg

consistently outperforms both EXP_temp and EXP_bg for the entire 3-h forecast period, which

implies that the integrated hydrometeor retrieval scheme is conducive to the assimilation of radar

reflectivity observations.



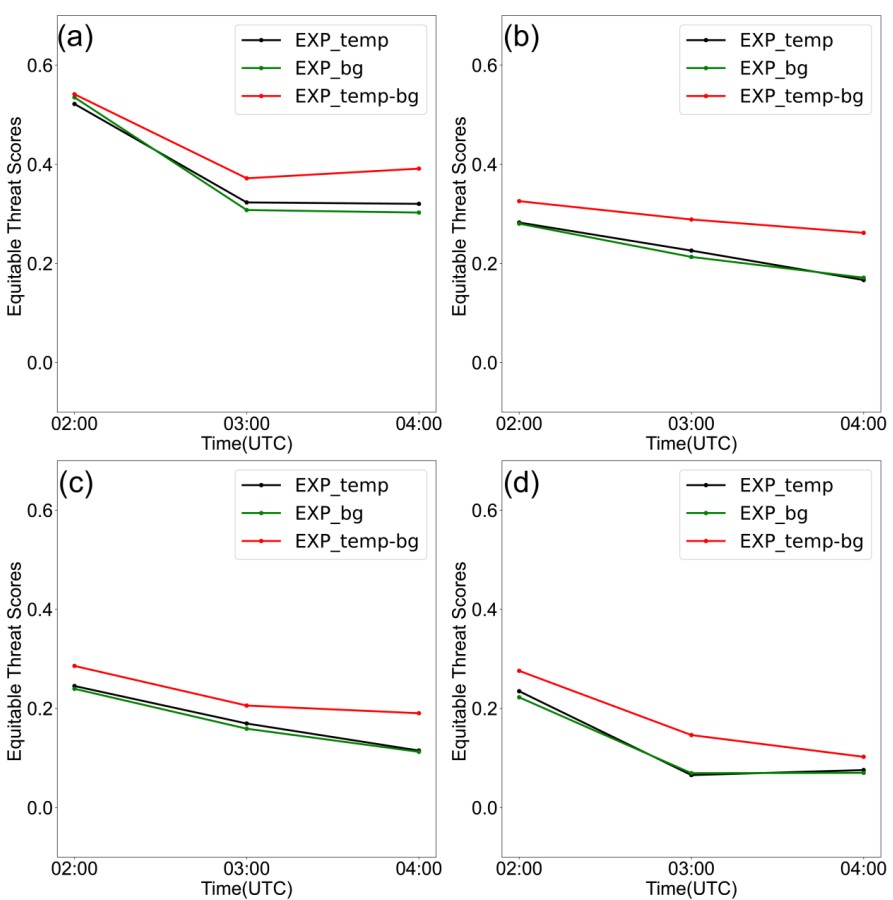


Fig. 8. Equitable threat scores of hourly accumulated precipitation forecasts with five thresholds: (a) 0.1 mm, (b)
2.5 mm, (c) 5 mm and (d) 10 mm from 2300 UTC 14 June to 0100 UTC 15 June.

## 4.2 06 August 2018 case

Fig. 9 presents the observed composite reflectivity at 1800UTC, 1900UTC, 2000UTC, and
2100UTC on 06 August 2018. At 1800 UTC, there are a small number of strong radar echoes in the
central part of Liaoning Province. At 1900UTC, these discrete strong echoes gradually converge in
the center Liaoning, forming a well-organized structure. By 2000UTC, the convections continue to
develop and form into "V" pattern echo. At 2100UTC, a distinct "T" shaped echo emerges in the
observed area.

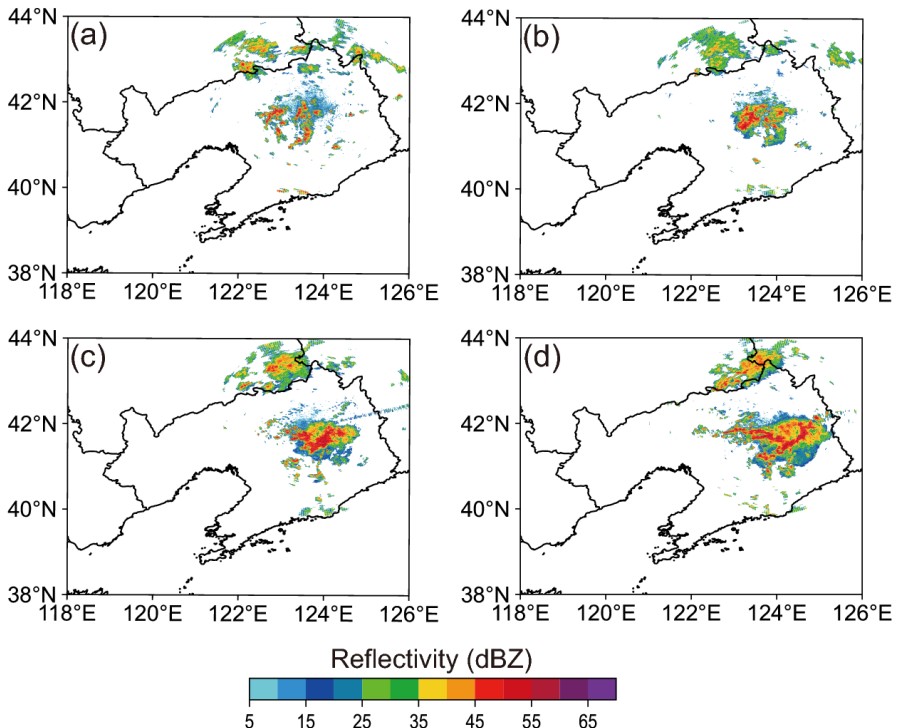


Fig. 9. The observed composite reflectivity fields (units: dBZ) at (a) 1800UTC, (b) 1900UTC, (c) 2000UTC and

(d) 2100UTC 06 August 2018.

Fig. 10 shows the radar reflectivity analysis fields and the vertical cross sections along the line

ab from EXP_temp, EXP_bg, and EXP_temp-bg at 2100 UTC. As shown in Fig. 10a, a distinct "T"

shaped echo emerges in the observed area. Generally, the composite reflectivity analyses of the

experiments EXP_temp, EXP_bg, and EXP_temp-bg show a general agreement. From the observed

vertical cross section, it seems that there exist three strong echo bands between 123.78°E and

124.36°E. In order to display the differences between three DA experiments and the observation,

the convective system located near 123.75°E is marked as A, the strong convection at 123.97°E -

124.17°E is named as B, and the strong echo region at 124.17°E -124. 36°E is labelled as C. Notably,

part A in the experiment EXP_temp departs from the observation, while EXP_bg and EXP_temp-

bg capture it more closely. Furtherly, the strong echo band analyzed by EXP_temp-bg indicates a

wider coverage than the one obtained from EXP_bg in part A. For part B, though all three DA

experiments exhibit a general agreement with the observation, their intensity is weaker than that in

the observation. It is found that EXP_temp-bg analyzes a strong center with reflectivity values

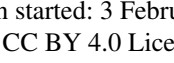


greater than 45dBZ for part B. All three experiments capture the overall structure of C. It seems
EXP_temp-bg combines the echo characteristics of both EXP_temp and EXP_bg in part C. On the
whole, EXP_temp-bg displays the advantages of fusion for most situations, matching best with the
observations.

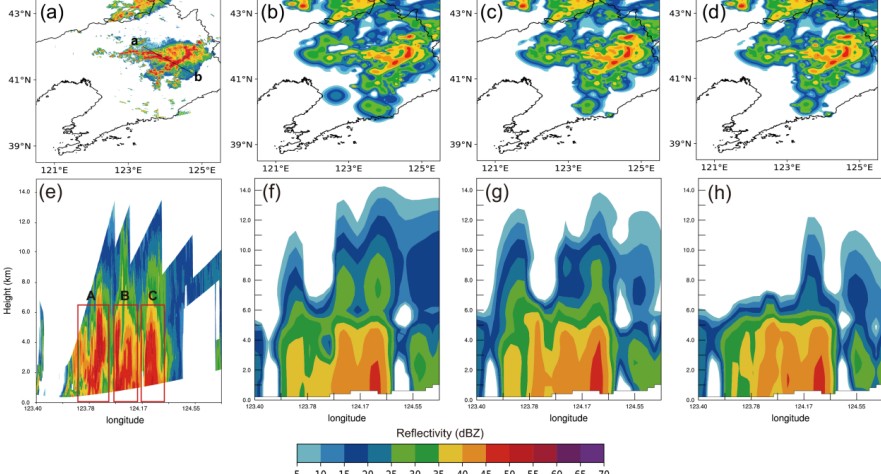


Fig. 10. The composite reflectivity at 2100 UTC for (a) observation, (b) EXP_temp, (c) EXP_bg, (d) EXP_temp-
bg, accompanied by the vertical cross sections for (e) observation, (f) EXP_temp, (g) EXP_bg, (h) EXP_temp-bg
along the line ab. The vertical cross section location at 2100UTC is shown by the line ab in the Fig. 10a. The labels

in the Fig. 10e present the convection locations.

To examine how different retrieval methods modify the hydrometeor distributions, the rainwater,

snow and graupel mixing ratio cross sections are presented in Fig. 11. Rainwater occurs below the
freezing level, while snow and graupel particles primarily exist above the freezing level. The
distribution of low-level rainwater in EXP_temp-bg is similar to that in EXP_bg. The proportion of
snow and graupel is a fixed coefficient in the EXP_temp, resulting in similar vertical distributions
as shown in Fig. 11a. However, it does not exist in the other two experiments with the "background
hydrometer-dependent" scheme. Additionally, both EXP_bg and EXP_temp-bg have significantly
higher snow and graupel content than EXP_temp. Fig. 11 shows three strong centers of graupel
particles corresponding to three strong reflectivity bands in the Fig. 10. By comparing the three
groups of the DA experiments, it is apparent that EXP_bg has the highest strong-center value, while
EXP_temp has the lowest. Moreover, the distribution of high-altitude hydrometeors in EXP_temp-



bg combines the features of EXP_temp and EXP_bg. To conclude, the hydrometeor vertical
distributions are closely related to the radar reflectivity structure as expected.

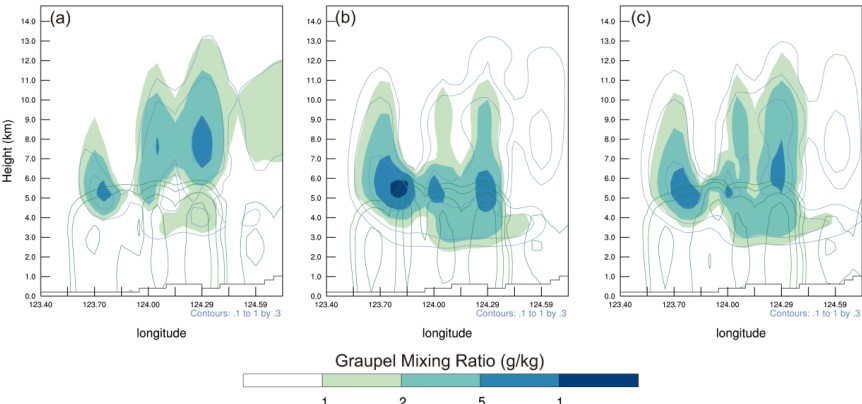


Fig. 11. The vertical cross sections of rainwater mixing ratio (green contours), snow mixing ratio (blue contours),
graupel mixing ratio (shading) at 2100 UTC for the experiments (a) EXP_temp, (b) EXP_bg, (c) EXP_temp-bg.
Fig. 12 displays the vertical cross sections of the pseudo-equivalent potential temperature (θse),
wind components, and reflectivity at 2100 UTC for EXP_temp, EXP_bg, and EXP_temp-bg. In the
three DA experiments, there exists a high θse zone in the lower layers (below 3 km), which shows
that a certain amount of energy has accumulated near the ground level. The area between 3 and 9
kilometers is characterized by a low θse zone, with the lowest value being below 343 K. Another
high θse zone exists above 10 kilometers. The results suggest that the vertical structure of the
atmosphere is unstable in this region, with dry conditions prevailing in the upper levels and moist
conditions in the lower levels. This type of vertical structure is favorable for the development of
severe convective weather events.
In the middle layer, there is a zone with relatively high θse value for EXP_bg and EXP_temp-bg.
Specifically, a warm-core structure is identified near 123.85°N, accompanied by strong upward
motion. This results in the release of unstable energy indicate that a severe convective system is
continuously developing. Additionally, compared with EXP_bg, EXP_temp-bg yields a more
extensive and deeper updraft column.

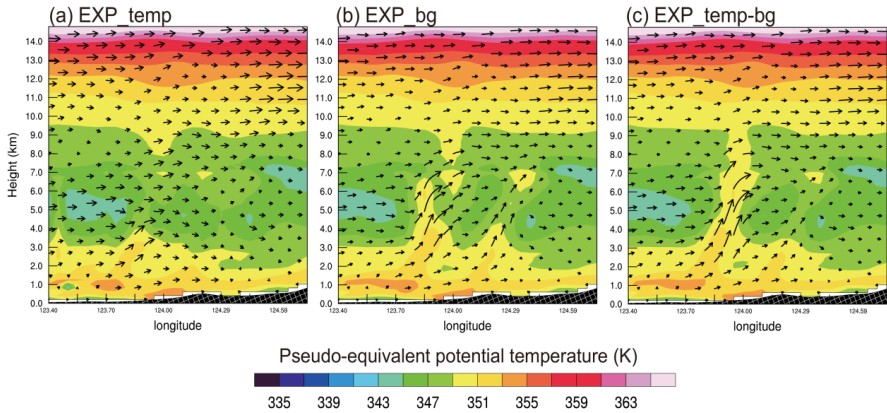

Fig. 12. The vertical sections of pseudo-equivalent potential temperature (shaded; units: K), velocity vectors (U,

W) at 2100 UTC for (a) EXP_temp, (b) EXP_bg and (c) EXP_temp-bg.

Fig. 13 shows 1-h, 3-h, and 5-h forecasts valid at 2100 UTC 06 August 2018 for EXP_temp,

EXP_bg, and EXP_temp-bg. As can be seen from the observation, the strong echo is located near

42°N at the beginning and has a tendency to slowly develop to the east. For the sake of clarity, the

strong echo zone is divided into two parts: part A and part B. At 2200 UTC 06 August, the forecasts

of three DA experiments for part B are inconsistent with the observation in terms of the intensity.

The part A predicted by EXP_bg and EXP_temp-bg shows a general agreement with the observation,

while the radar reflectivity forecast of EXP_temp departs from the observation. At 0000 UTC 07

August, EXP_bg and EXP_temp-bg yield an improved forecast for part A and B as compared with

EXP_temp, in terms of the intensity and organization. However, there is a southeast bias in part A

predicted by both EXP_bg and EXP_temp-bg. Compared to EXP_bg, EXP_temp-bg provides more

accurate predictions for part B. As shown by the observation at 0200 UTC 07 August, the predicted

A in EXP_temp-bg shows closer alignment with the observation than that in EXP_temp and

EXP_bg. For part B, three sets of experiments all depart from the observation. Overall, EXP_temp-

bg demonstrates superior prediction skills in terms of the radar reflectivity.

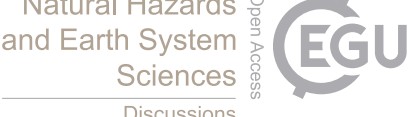

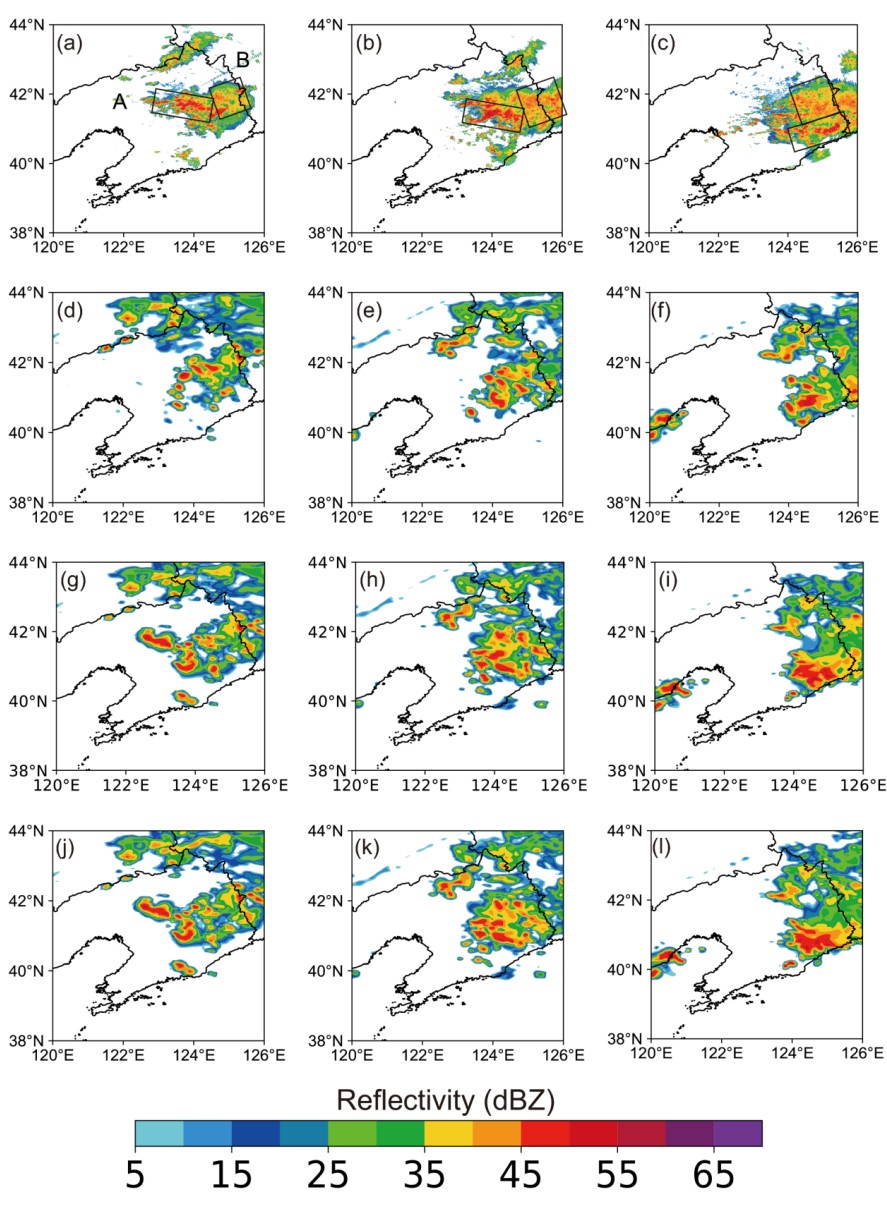


Fig. 13. The composite reflectivity (shaded; units: dBZ) predicted by (d)-(f) EXP_temp (g)-(i) EXP_bg and (j)-(l)
EXP_temp-bg, as compared to (a)-(c) the observed composite reflectivity. The corresponding times from left to
right are 2200 UTC 06 August (left), 0000 UTC 07 August (middle) and at 0200 UTC 07 August (right),
respectively. The labels A and B present the convection locations.
Fig. 14 shows 6-h accumulated precipitation of the three DA experiments from 2100 UTC 06


364 August to 0300 UTC 07 August 2018. According to the observation, heavy rainfall is mainly

365 concentrated in the northeastern part of Liaoning, with precipitation amount exceeding 100 mm. All

366 three experiments underestimate the extent of the precipitation in this event, especially in the range

367 of 25 mm to 50 mm. Moreover, there is a certain deviation between the predicted and observed

368 locations. As shown in Fig. 9c and d, the patterns of heavy precipitation areas are similar in EXP_bg

369 and EXP_temp-bg. EXP_bg and EXP_temp-bg are notably better than EXP_temp in predicting the

370 rainfall for the threshold 50mm. EXP_temp-bg displays the best forecasting skill in terms of the

371 heavy rainfall area.

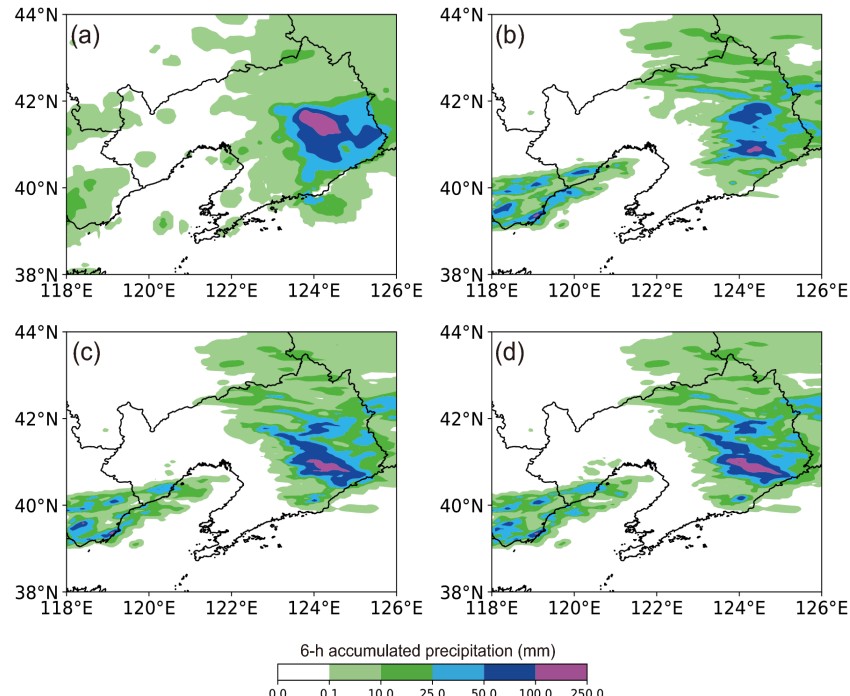

373 Fig. 14. 6-h accumulated precipitation valid at 2100 UTC 06 August 2018. (a) the observation, (b) EXP_temp, (c)

374 EXP_bg, and (d) EXP_temp-bg.

## 5. The conclusion

376 The study proposes an adaptive hydrometeor retrieval scheme within the WRF-3DVar system,

377 which combines "temperature-based" and "background hydrometer-dependent" methods to

378 enhance the analyses and forecasts for the strong convections. In the indirect assimilation of radar





reflectivity, it is vital to correctly divide hydrometeor information in radar reflectivity. On the basis
of two retrieval methods proposed by Gao and Stensrud (2012) and Chen et al. (2020, 2021), the
blending scheme is developed to minimize the limitations brought by both methods so as to improve
the assimilation and prediction skills.
The above three hydrometeor retrieval schemes are evaluated for two strong convective processes
occurred during June 2020 and August 2018. Three DA experiments (EXP_temp, EXP_bg, and
EXP_temp-bg) are conducted by using the "temperature-based", "background hydrometer-
dependent", and blending methods, respectively. The analysis results reveal that the blending
method is effective to improve the radar reflectivity structures for severe convections. Based on the
other two DA experiments, EXP_temp-bg further improves hydrometeor structures and properly
allocates the proportion of each hydrometeor, which is responsible for more reasonable hydrometeor
distributions. Also, EXP_temp-bg provides more reasonable dynamic and thermal structures
compared with EXP_temp and EXP_bg. EXP_temp-bg shows advantages in the precipitation
prediction skills due to the reasonable spatial distribution and proportion of each hydrometeor.
Compared to conventional Doppler weather radars, dual-polarization radar observations provide
more accurate identification of the three-dimensional microphysical structures within precipitation
systems. Consequently, dual-polarization radar data will be considered for hydrometeor retrievals
in our future studies, aiming to further enhance the forecast skills for severe weathers.

**Data availability**
The GFS reanalysis data is available at https://rda.ucar.edu/datasets/ds084.1/, and the source code
of WRF and WRFDA can be downloaded from https://github.com/wrf-model. The radar
observations after quality control are provided by Jiangsu and Liaoning Provincial Meteorological
Bureau, and the precipitation observations can be found at
http://data.cma.cn/dataService/cdcindex/datacode/NAFP_CLDAS2.0_NRT/show_value/normal.ht
ml.

**Author contribution**
LS: visualization, writing (original draft). FS: conceptualization, writing (review and editing). ZH:



conceptualization, methodology. DX: writing (review and editing). AS: visualization. JC: software.

**Competing interests**
The contact author has declared that none of the authors has any competing interests.

*Acknowledgments*

This research was primarily supported by the Chinese National Natural Science Foundation of China

(G42192553), the China Meteorological Administration Tornado Key Laboratory (TKL202306), Natural
Science Fund of Anhui Province of China under grant (2308085MD127), the Open Grants of China
Meteorological Administration Radar Meteorology Key Laboratory (2023LRM-B03), the Open Project
Fund of China Meteorological Administration Basin Heavy Rainfall Key Laboratory (2023BHR-Y20),
the Shanghai Typhoon Research Foundation (TFJJ202107), the Chinese National Natural Science
Foundation of China (G41805070). We acknowledge the High Performance Computing Center of
Nanjing University of Information Science & Technology for their support of this work.

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
