# Peer review of "Indirect assimilation of radar reflectivity data with an adaptive hydrometer retrieval scheme for the short-term severe weather forecasts"

_Natural Hazards and Earth System Sciences, 2024_

## Author Response (AR1)

**Reply to referee (1)'s comments on NHESS-2024-203:**

We would like to thank the reviewer for the invaluable comments and suggestions. Here are our responses to the reviewer's comments.

**Major comments**

1. In case 2, the author did not conduct a quantitative evaluation of radar reflectivity or precipitation forecasts. A more thorough quantitative assessment could be provided to better validate the performances of the different retrieval schemes.

Reply: Thanks for the valuable comments. Added as "Fig. 15 shows ETS values of 1-h accumulated precipitation for EXP\_temp, EXP\_bg, and EXP\_temp-bg. For the thresholds of 2.5 mm/h, the precipitation forecasts of EXP\_temp-bg generally exhibit superior quality. The experiment EXP\_temp keeps the worst for the ETS scores among the three sets of experiments. At thresholds of 10 mm/h, the score of EXP\_temp-bg gradually increases in the later stage of forecast. The scores indicat that the blending method is able to improve the precipitation forecast skill." in the section 4.2.

Fig. 15. ETS of three DA experiments for the thresholds of (a) 2.5 mm/h, (b) 10mm/h.

2. The description of the background hydrometer-dependent method is not clear, particularly regarding key implementation details. For example, it is unclear how the radar reflectivity threshold intervals are defined, what sample size is used for the background statistics, and how the climatological data are calculated. Providing more details in these aspects can help to enhance the reproducibility and transparency of the method.

Reply: Thanks for the valuable comments. Added as "It is found that hydrometeor weights derived from the background field vary with individual weather conditions, which helps to reduce errors resulting from fixed coefficients in Chen et al. (2020, 2021). The specific process of calculating proportions is as follows:

- (1) Compute the average equivalent radar reflectivity of each hydrometeor  $(\overline{Z}_{x_{(k,ref_1)}})$  in different reflectivity ranges  $(ref_i)$  and model layers (k) based on the background field statistics. The reflectivity ranges are usually set as follows:  $ref_1 < 15 \text{ dBZ}$ ,  $15 \text{ dBZ} \le ref_2 < 25 \text{ dBZ}$ ,  $25 \text{ dBZ} \le ref_3 < 35 \text{ dBZ}$ ,  $35 \text{ dBZ} \le ref_4 < 45 \text{ dBZ}$ ,  $ref_5 \ge 45 \text{ dBZ}$ .
- (2) Calculate the weight  $(C_{x_{(k,ref_i)}})$  of each hydrometeor in the background field.

$$C_{x_{(k,ref_i)}} = \overline{Z_{x_{(k,ref_i)}}} / \overline{Z_{total_{(k,ref_i)}}}, \tag{9}$$

$$\overline{Z_{total_{(k,ref_1)}}} = \overline{Z_{r_{(k,ref_1)}}} + \overline{Z_{s_{(k,ref_1)}}} + \overline{Z_{g_{(k,ref_1)}}}.$$
(10)

- (3) Divide radar reflectivity observations based on the weights  $(C_{x_{(k,ref_i)}})$  derived from Step 2. If the background field has missing data, the calculated climatological mean for one month will be used instead." in the section 2.3.2
- 3. The last paragraph of the introduction provides only a superficial listing of each section's content. To improve the clarity and effectiveness of the paper, it is recommended to expand on the role of each section. Elaborate on how they contribute to the overall narrative and objectives of the research, which will help readers gain a clearer understanding of the study's scope and significance.

Reply: Thanks for the valuable comments. "In the study, section 2 describes the WRF-3DVar methods, radar observation operators, and a new hydrometeor retrieval method that adaptively combines the "temperature-based" and "background hydrometeor-dependent" methods. Based on two convective cases, three experiments are designed to investigate the impact of different hydrometeor retrieval schemes on assimilation and prediction, with the specific configurations presented in section 3. The section 4 presents analysis and forecast results of all experiments. The conclusion and is presented in the section 5."

4. The writing needs further improvement. It is recommended that the authors engage a professional editor or a native English speaker to proofread it, which would significantly boost the clarity and coherence of the manuscript.

Reply: Thanks for the valuable comments. The manuscript has been thoroughly revised to improve its writing and coherence.

**Minor comments**

 Abstract: The abstract does not explicitly address the impact of thermodynamic and dynamic structures on the forecast results. Given that the evolution of convective systems is closely related to environmental thermodynamics and dynamics (e.g., vertical velocity and wind shear), including a brief statement on how the proposed method enhances key thermodynamic structures would make the abstract more comprehensive.

Reply: Thanks for the valuable comments. The abstract is revised as "Different hydrometeor retrieval schemes are explored based on the Weather Research and Forecasting (WRF) model in the indirect assimilation of radar reflectivity for two real cases occurred during June 2020 and August 2018. When retrieving hydrometeors from radar reflectivity, there are two commonly used hydrometeor classification methods: "temperature-based" and "background hydrometer-dependent" schemes. The hydrometeor proportions are usually empirically assigned in the "temperature-based" method within different background temperature intervals. Whereas, in the "background hydrometer-dependent" scheme, each type of the hydrometeor is derived based on the portions estimated from the background field for different radar reflectivity ranges. In this study, a blending scheme is designed to combine "temperature-based" and "background hydrometer-dependent" methods adaptively to avoid errors caused by fixed relationships and reduce uncertainties introduced by the background field itself. Three experiments, EXP\_temp, EXP\_bg, and EXP\_temp-bg are conducted using the "temperature-based" method, "background hydrometer-dependent" scheme, and blending scheme, respectively. It is found that, adding the "background hydrometerdependent" scheme facilitates the generation of accurate hydrometeor species which will enhance the effectiveness of radar data assimilation. Besides, due to the adaptive combination of "temperature-based" and "background hydrometer-dependent" schemes, the EXP\_temp-bg experiment yields the improved thermodynamic and dynamic structures, which contributes to predict radar reflectivity and precipitation intensity more accurately."

2. Line 48: Please rephrase this sentence "the EXP\_temp-bg experiment predict the radar reflectivity structures and precipitation intensity more accurately"

Line 254: spelling mistake: "produces" → "produces"

Line 344: "1-h, 3-h, and 5-h forecasts valid at 2100 UTC 06 August 2018 for EXP\_temp"?

Reply: Thanks. The sentences are revised as "the EXP\_temp-bg experiment predicts the radar reflectivity structures and precipitation intensity more accurately."

"In comparison, EXP\_temp-bg produces the most consistent thermal and dynamical conditions, resulting in most accurate forecast of the convection."

"6-h accumulated precipitation initialized at 2100 UTC 06 August 2018."

 Section 2: Please use the style requirements of American Meteorological Society uniformly in words, formulas and charts. For example, single-character variables should be italicized; Use non-italic bold for vectors or matrices.

Reply: Thanks. The related formulae have been modified.

Based on the incremental method proposed by Courtier et al. (1994), 3DVar uses the minimization algorithm to solve the objective function. The cost function is as follows:

$$J = \frac{1}{2} (\mathbf{x} - \mathbf{x}_{b})^{\mathrm{T}} \mathbf{B}^{-1} (\mathbf{x} - \mathbf{x}_{b}) + \frac{1}{2} [H(\mathbf{x}) - \mathbf{y}_{o}]^{\mathrm{T}} \mathbf{R}^{-1} [H(\mathbf{x}) - \mathbf{y}_{o}].$$
(1)

The vectors  $\mathbf{x}$ ,  $\mathbf{x}_{b}$  and  $\mathbf{y}_{o}$  stand for analysis variables, background variables, and observation variables.  $\mathbf{B}$  is the background error covariance, which is calculated by the National Meteorological Center (NMC; Parrish and Derber, 1992) method.  $\mathbf{R}$  represents the observation error covariance. H is the nonlinear observation operator.

$$V_r = u \frac{x - x_i}{r_i} + v \frac{y - y_i}{r_i} + (w - v_T) \frac{z - z_i}{r_i}.$$
 (2)

u, v, and w denote the zonal, meridional, and vertical wind component, respectively. (x, y, z) and  $(x_i, y_i, z_i)$  represent the radar position and observation position,

respectively.  $r_i$  is the distance between the radar and the observation.  $v_T$  is the terminal speed.

$$Z = 10 * \log_{10}(Z_{\rho}), \tag{3}$$

$$Z_e = Z_e(q_r) + Z_e(q_s) + Z_e(q_g),$$
 (4)

$$Z_e(q_x) = \alpha_x (\rho q_x)^{1.75}. \tag{5}$$

 $q_x$  means hydrometeor mixing ratios.  $Z_e(q_x)$  (units: dBZ) is the equivalent reflectivity factor of rainwater, snow, and graupel.  $\alpha_x$  represents the fixed coefficient that is determined by the dielectric coefficient, density and intercept parameter of each hydrometeor.  $\alpha_r$  is  $3.63\times10^9$ . For snow and graupel, the coefficient is temperature dependent. When the environmental temperature is greater than  $0^{\circ}$ C,  $\alpha_s$  for wet snow is  $4.26\times10^{11}$  and  $\alpha_g$  for wet graupel is  $9.08\times10^9$ . When the temperature is below  $0^{\circ}$ C,  $\alpha_s$  for dry snow is  $9.80\times10^8$  and  $\alpha_g$  for dry graupel is  $1.09\times10^9$ .  $\rho$  is the air density. During the direct assimilation of radar reflectivity, the linearization errors are almost inevitable.

**4. Section 2.3 L148~154: Does $\alpha$ and a represent the same variable?**

Reply: We thank the reviewer for the valuable comment. It is a typo, and we have corrected it as follows.

$$Z = 10 * \log_{10}(Z_e), \tag{3}$$

$$Z_e = Z_e(q_r) + Z_e(q_s) + Z_e(q_g),$$
 (4)

$$Z_e(q_x) = \alpha_x (\rho q_x)^{1.75}. (5)$$

 $q_x$  means hydrometeor mixing ratios.  $Z_e(q_x)$  (units: dBZ) is the equivalent reflectivity factor of rainwater, snow, and graupel.  $\alpha_x$  represents the fixed coefficient that is determined by the dielectric coefficient, density and intercept parameter of each hydrometeor.  $\alpha_r$  is  $3.63\times10^9$ . For snow and graupel, the coefficient is temperature dependent. When the environmental temperature is greater than  $0^{\circ}$ C,  $\alpha_s$  for wet snow is  $4.26\times10^{11}$  and  $\alpha_g$  for wet graupel is  $9.08\times10^9$ . When the temperature is below  $0^{\circ}$ C,  $\alpha_s$  for dry snow is  $9.80\times10^8$  and  $\alpha_g$  for dry graupel is  $1.09\times10^9$ .  $\rho$  is the air density. During the direct assimilation of radar reflectivity, the linearization errors are almost inevitable.

5. Section 2.3.3: Equations (9) and (10) defining the blending scheme lack a clear explanation of how the weighting factors are determined.

Reply: We appreciate the reviewer's suggestion.

Added as "2.3.2 The "Background hydrometeor-dependent" method

It is found that hydrometeor weights derived from the background field vary with individual weather conditions, which helps to reduce errors resulting from fixed coefficients in Chen et al. (2020, 2021). The specific process of calculating proportions is as follows:

- (1) Compute the average equivalent radar reflectivity of each hydrometeor  $(\overline{Z}_{x(k,ref_l)})$  in different reflectivity ranges  $(ref_l)$  and model layers (k) based on the background field statistics. The reflectivity ranges are usually set as follows:  $ref_1 < 15$  dBZ, 15 dBZ  $\leq ref_2 < 25$  dBZ, 25 dBZ  $\leq ref_3 < 35$  dBZ, 35 dBZ  $\leq ref_4 < 45$  dBZ,  $ref_5 \geq 45$  dBZ.
- (2) Calculate the weight  $(C_{x_{(k,ref_i)}})$  of each hydrometeor in the background field.

$$\overline{Z_{total_{(k,ref_l)}}} = \overline{Z_{r_{(k,ref_l)}}} + \overline{Z_{s_{(k,ref_l)}}} + \overline{Z_{g_{(k,ref_l)}}}.$$
 (10)

(3) Divide radar reflectivity observations based on the weights  $(C_{x_{(k,ref_i)}})$  derived from Step 2. If the background field has missing data, the calculated climatological mean for one month will be used instead.

**2.3.3 The blending method**

The blending method aims to utilize the two methods of partitioning hydrometeors accordingly to retrieve muti-hydrometer more reasonably in radar reflectivity indirect assimilation. Firstly, calculate the standard deviation  $\sigma$  of each hydrometeor content in the model grid and its surrounding background grids. If the standard deviations of the retrieved hydrometeors of the two schemes are less than  $2\sigma$ , it means that the retrieved hydrometeors are consistent with the local structure of the background. Therefore, the hydrometeor content is calculated by the following formulas:

$$\beta = \frac{\delta_t^2}{\delta_t^2 + \delta_b^2},\tag{11}$$

$$C_x = \beta C_x^b + (1 - \beta)C_x^t. \tag{12}$$

 $\delta_t^2$  represents the deviation between the hydrometeor content of the background field and the retrieved hydrometeor content based on the "temperature-based" scheme.  $\delta_b^2$  is the deviation between the hydrometeor content of the background field and the retrieved hydrometeor by the "background hydrometer-dependent" scheme.  $C_x^t$  and  $C_x^b$  are the weights calculated by the "temperature-based" and "background hydrometer-dependent" methods, respectively.  $\beta$  means the proportion of the results calculated by "background hydrometer-dependent" method." In the section 2.

6. Section 2.3.3: What is the weight meaning  $\beta$  for? It needs more description.

Reply: Thanks. Added as "The blending method aims to utilize the two methods of partitioning hydrometeors accordingly to retrieve muti-hydrometer more reasonably in radar reflectivity indirect assimilation. Firstly, calculate the standard deviation  $\sigma$  of each hydrometeor content in the model grid and its surrounding background grids. If the standard deviations of the retrieved hydrometeors of the two schemes are less than  $2\sigma$ , it means that the retrieved hydrometeors are consistent with the local structure of the background. Therefore, the hydrometeor content is calculated by the following formulas:

$$\beta = \frac{\delta_t^2}{\delta_t^2 + \delta_b^2},\tag{11}$$

$$C_x = \beta C_x^b + (1 - \beta)C_x^t. \tag{12}$$

 $\delta_t^2$  represents the deviation between the hydrometeor content of the background field and the retrieved hydrometeor content based on the "temperature-based" scheme.  $\delta_b^2$  is the deviation between the hydrometeor content of the background field and the retrieved hydrometeor by the "background hydrometer-dependent" scheme.  $C_x^t$  and  $C_x^b$  are the weights calculated by the "temperature-based" and "background hydrometer-dependent" methods, respectively.  $\beta$  means the proportion of the results calculated by "background hydrometer-dependent" method. "In the section 2.

7. Section 3: The paper mentions the use of radar observations but does not provide sufficient details on the quality control procedures.

Reply: Thanks for the valuable comments. Added as "The radar observations of two cases undergo a series of preprocessing and quality control procedure, including

anomaly detection, velocity de-aliasing, and so on" in the section 3.

8. Section 3: The observation error statistics estimated and used in DA determine the increment field for given innovations. Quantitative details of these statistics are crucial for understanding DA results but are not provided. Also, a more detailed description of the experiment design is required.

Reply: Thanks for the valuable comments. Added as "WRF v4.3 and its data assimilation system WRFDA v4.3 are used in this study. Two convective cases are studied in the study: 14 June in 2020 (called Case 1; Fig. 1a) and 6 August in 2018 (denoted as Case 2; Fig. 1b). For case 1, the model domain contains 500×471 with a 3 km horizontal grid spacing, and 50 vertical levels. For case 2, the model domain contains 723 × 691 with a 3 km horizontal grid spacing, and 50 vertical levels. The specific applications of physical parametrizations are as follows: the WRF Double-Moment 6-Class Microphysics (WDM6) scheme, the Rapid Radiative Transfer Model (RRTM) long wave radiation scheme (Mlawer et al., 1997), the Dudhia short-wave radiation scheme (Dudhia, 1989), the Yonsei University (YSU) boundary layer scheme (Hong et al., 2006), and the Noah Land Surface Model (Chen and Dudhia, 2001) for land surface process scheme. No cumulus parameterization scheme is employed. As shown in Table 1, three data assimilation (DA) experiments are conducted to evaluate the effects of all retrieval methods in the study. For all three DA experiments, the initial and lateral boundary conditions are provided by the NCEP Global Forecast System (GFS) data. Besides, the specific flowchart is presented in the Fig. 2. The radar observations used in two cases undergo a series of preprocessing and quality control procedure, including anomaly detection, velocity de-aliasing, and so on. The observation errors of radar radial velocity and radar reflectivity are set to 2 m s-1 and 5 dBZ, respectively." in the section 3.

9. Section 4: In the Fig. 6 and Fig. 12, the wind speed scale needs to be given in the lower right corner of the figure, and the length of the wind vector indicates the wind speed, and the unit is how much.

Reply: Thanks. The pictures have been revised.

Fig. 12. The vertical sections of pseudo-equivalent potential temperature (shaded; units: K), velocity vectors (units: m/s; the vertical velocity has been multiplied by 10) at 2100 UTC for (a) EXP\_temp, (b) EXP\_bg and (c) EXP\_temp-bg. The position of the cross sections is located at the line ab of the Fig. 10a.

10. Section 4.2: Where is the cross section shown in Figure 12?Reply: Thanks for the valuable comments. It is added in the caption of Fig. 12.

Fig. 12. The vertical sections of pseudo-equivalent potential temperature (shaded; units: K), velocity vectors (units: m/s; the vertical velocity has been multiplied by 10) at 2100 UTC for (a) EXP\_temp, (b) EXP\_bg and (c) EXP\_temp-bg. The position of the cross sections is located at the line ab of the Fig. 10a.

11. Section 5: The conclusion mentions using dual-polarization radar in future studies, but does not elaborate on how this would be integrated into the current framework. Providing more details on potential improvements or challenges would strengthen the future outlook.

Reply: Thanks. It is revised with "Compared to conventional Doppler weather radars, dual-polarization radar observations provide more accurate identification of the three-dimensional microphysical structures within precipitation systems. Consequently, dual-polarization radar data (e.g. differential reflectivity, specific differential phase, correlation coefficient) will be considered for identifying the hydrometeor types more accurately, aiming to enhance the effectiveness of radar data assimilation."

12. Some pictures (e.g. Fig. 4, Fig. 5, Fig. 10, Fig. 11, Fig. 12) are too small to read clearly. Please enlarge the labels for better visibility.

Reply: Thanks for the valuable comments. All the pictures in the manuscript have been re-examined and revised.

Fig. 4. The vertical sections of (a) hydrometeor classification algorithm based on the dual-polarization radar observations and retrieved hydrometeors for (b) EXP\_temp, (c) EXP\_bg and (d) EXP\_temp-bg along the black lines a1-a2 at 1500 UTC. The retrieved hydrometeors refer to rainwater mixing ratio (green contours; units: dBZ), dry snow mixing ratio (grey contours; units: dBZ), wet snow mixing ratio (cyan contours; units: dBZ), and graupel mixing ratio (shading; units: dBZ), respectively.

Fig. 5. The composite reflectivity (shaded; units: dBZ) predicted by (e)-(h) EXP\_temp (i)-(l) EXP\_bg and (m)-(p) EXP\_temp-bg for the 1-h forecast beginning at 0100 UTC 15 June 2020, as compared to (a)-(d) the observed composite reflectivity. The labels C and D present the convection locations.

Fig. 7. 3-h accumulated precipitation initialized at 0100 UTC 15 June 2020. (a) the observation, (b) EXP\_temp, (c) EXP\_bg, and (d) EXP\_temp-bg.

Fig. 10. The composite reflectivity at 2100 UTC for (a) observation, (b) EXP\_temp, (c) EXP\_bg,

(d) EXP\_temp-bg, accompanied by the vertical cross sections for (e) observation, (f) EXP\_temp, (g) EXP\_bg, (h) EXP\_temp-bg along the line ab. The vertical cross section location at 2100UTC is shown by the line ab in the Fig. 10a. The labels in the Fig. 10e present the convection locations.

Fig. 11. The vertical cross sections of rainwater mixing ratio (green contours), snow mixing ratio (blue contours), graupel mixing ratio (shading) at 2100 UTC for the experiments (a) EXP\_temp, (b) EXP\_bg, (c) EXP\_temp-bg. The position of the cross sections is located at the line ab of the Fig. 10a.

Fig. 12. The vertical sections of pseudo-equivalent potential temperature (shaded; units: K), velocity vectors (units: m/s; the vertical velocity has been multiplied by 10) at 2100 UTC for (a) EXP\_temp, (b) EXP\_bg and (c) EXP\_temp-bg. The position of the cross sections is located at the line ab of the Fig. 10a.

Reply to referee (2)'s comments on NHESS-2024-203:

We would like to thank the reviewer for the invaluable comments and suggestions. Here are our responses to the referee's comments.

**Major comments**

The study proposes an adaptive hydrometeor retrieval scheme, which adaptively combines "temperature-based" and "background hydrometer-dependent" methods to improve analyses and forecasts of two real cases occurred during June 2020 and August 2018. However, the manuscript lacks more details about the retrieval scheme itself. Specifically, it would be helpful to provide a more detailed explanation of how the two methods are combined, including the criteria or algorithm used for their weighting. Additionally, English writing needs to be further improved.

Reply: Thanks. It is revised with "

**2.3.2 The "Background hydrometeor-dependent" method**

It is found that hydrometeor weights derived from the background field vary with individual weather conditions, which helps to reduce errors resulting from fixed coefficients in Chen et al. (2020, 2021). The specific process of calculating proportions is as follows:

- (1) Compute the average equivalent radar reflectivity of each hydrometeor  $(\overline{Z}_{x_{(k,ref_i)}})$  in different reflectivity ranges  $(ref_i)$  and model layers (k) based on the background field statistics. The reflectivity ranges are usually set as follows:  $ref_1 < 15 \text{ dBZ}$ ,  $15 \text{ dBZ} \le ref_2 < 25 \text{ dBZ}$ ,  $25 \text{ dBZ} \le ref_3 < 35 \text{ dBZ}$ ,  $35 \text{ dBZ} \le ref_4 < 45 \text{ dBZ}$ ,  $ref_5 \ge 45 \text{ dBZ}$ .
- (2) Calculate the weight  $(C_{x_{(k,ref_i)}})$  of each hydrometeor in the background field.

$$\overline{Z_{total_{(k,ref_t)}}} = \overline{Z_{r_{(k,ref_t)}}} + \overline{Z_{s_{(k,ref_t)}}} + \overline{Z_{g_{(k,ref_t)}}}.$$
(10)

(3) Divide radar reflectivity observations based on the weights  $(C_{x_{(k,ref_i)}})$  derived from Step 2. If the background field has missing data, the calculated climatological mean for one month will be used instead.

**2.3.3 The blending method**

The blending method aims to utilize the two methods of partitioning hydrometeors accordingly to retrieve muti-hydrometer more reasonably in radar reflectivity indirect assimilation. Firstly, calculate the standard deviation  $\sigma$  of each hydrometeor content in the model grid and its

surrounding background grids. If the standard deviations of the retrieved hydrometeors of the two schemes are less than  $2\sigma$ , it means that the retrieved hydrometeors are consistent with the local structure of the background. Therefore, the hydrometeor content is calculated by the following formulas:

$$\beta = \frac{\delta_t^2}{\delta_t^2 + \delta_b^2},\tag{11}$$

$$C_x = \beta C_x^b + (1 - \beta) C_x^t. \tag{12}$$

 $\delta_t^2$  represents the deviation between the hydrometeor content of the background field and the retrieved hydrometeor content based on the "temperature-based" scheme.  $\delta_b^2$  is the deviation between the hydrometeor content of the background field and the retrieved hydrometeor by the "background hydrometer-dependent" scheme.  $C_x^t$  and  $C_x^b$  are the weights calculated by the "temperature-based" and "background hydrometer-dependent" methods, respectively.  $\beta$  means the proportion of the results calculated by "background hydrometer-dependent" method. "In the section 2. Besides, the writing has been further improved.

**Minor comments**

1. The manuscript lacks a detailed description of the used radars in the two case studies.

Reply: Thanks. Added as "Besides, the specific flowchart is presented in the Fig. 2. The radar observations used in two cases undergo a series of preprocessing and quality control procedure, including anomaly detection, velocity de-aliasing, and so on. The observation errors of radar radial velocity and radar reflectivity are set to 2 m s-1 and 5 dBZ, respectively." in the section 3.

Fig. 1. The simulated area of (a) Case 1 and (b) Case 2, with the detecting ranges of the Nanjing radar and Shenyang Radar. Both radars are S-band Doppler radars with a maximum coverage range of 230 km. The radial velocity and reflectivity observations have range resolutions of 250 m and 1000 m, respectively.

2. Due to the critical role of water vapor in the development of strong convection, it is necessary to clarify whether water vapor is assimilated and how it is assimilated in the method section.

Reply: Thanks.

Added as "Therefore, the indirect assimilation method is utilized in the study. The indirect method assimilates the retrieved water vapor and hydrometeors from the radar reflectivity observations. Following Wang et al. (2013), it is assumed that when the radar reflectivity exceeds a certain threshold, the relative humidity reaches 100%. The threshold is set to 30 dBZ in this study. The saturation water vapor at that point is then calculated and assimilated as a pseudo observation.

For retrieving hydrometeors from radar reflectivity, it is required to determine the proportion of each hydrometeor in radar reflectivity observation. At present, there are two methods to obtain the proportion of each hydrometeor." in the section 2.3.

3. Comparing Fig. 10 and Fig. 11, it seems that the vertical distribution of hydrometeor and radar reflectivity do not match.

Reply: Sorry, we misplaced the picture. The correct pictures are as follows.

Fig. 10. The composite reflectivity at 2100 UTC for (a) observation, (b) EXP\_temp, (c) EXP\_bg, (d) EXP\_temp-bg, accompanied by the vertical cross sections for (e) observation, (f) EXP\_temp, (g) EXP\_bg, (h) EXP\_temp-bg along the line ab. The vertical cross section location at 2100UTC is shown by the line ab in the Fig. 10a. The labels in the Fig. 10e present the convection locations.

4. In the second case study, the evaluation of all experiments is primarily qualitative, lacking a thorough quantitative assessment.

Reply: Thanks. Added as "Fig. 15 shows ETS values of 1-h accumulated precipitation for EXP\_temp, EXP\_bg, and EXP\_temp-bg. For the thresholds of 2.5 mm/h, the precipitation forecasts of EXP\_temp-bg generally exhibit superior quality. The experiment EXP\_temp keeps the worst for the ETS scores among the three sets of experiments. At thresholds of 10 mm/h, the score of EXP\_temp-bg gradually increases in the later stage of forecast. The scores indicat that the blending method is able to improve the precipitation forecast skill." in the section 4.2.

Fig. 15. ETS of three DA experiments for the thresholds of (a) 2.5 mm/h, (b) 10 mm/h.

5. Line 316, "However, it does not exist in the other two experiments ....." the sentence is not clearly expressed.

Reply: Thanks. It is modified for "For schemes associated with the background, the weights assigned to different hydrometeors vary dynamically with the background field. Therefore, the fixed coefficient does not exist in the other two experiments (EXP\_bg and EXP\_temp-bg)" in the section 4.2.

6. Line 368, "As shown in Fig. 9c and d"?

Reply: Thanks. It is modified for "As shown in Fig. 14c and d, the patterns of heavy precipitation areas are similar in EXP\_bg and EXP\_temp-bg."

7. It is beneficial to add a water vapor cross section in the second case similar to Figure 6.
Providing such a cross section would help explain the impact of the different retrieval scheme on moisture distribution.

Reply: Thanks. The figure displays the vertical profiles of the relative humidity, radar reflectivity and wind fields at 2200 UTC 6 August. Compared to EXP-temp and EXP-bg, EXP-temp\_bg indicate the presence of stronger reflectivity near the strong convection, accompanied by strong updrafts that facilitate vertical moisture transport from the lower to upper levels. In particular, EXP-temp\_bg shows a pronounced moisture column and reflectivity profile extending up to approximately 10 km.

Fig. The cross sections of relative humidity (shading; units: %), radar reflectivity (black contours starting at 40 dBZ; units: dBZ), and wind vectors for (a) EXP\_temp, (b) EXP\_bg and (c) EXP\_temp-bg along the longitude line of 123.6°E. These are 1-hour forecasts initialized at 2100 UTC. The blue rectangle at the bottom of the figure shows

the strong echo area.

We would like to thank the community for the invaluable comments and suggestions. Here are our responses to the community's comments.

The methods and experimental setup section would benefit from additional details. The specific implementation of the retrieval method is unclear and should be elaborated. Moreover, the method of statistical background error covariance needs further explanation. The experimental procedures for the two case studies also need to be described in greater detail.

Reply: Thanks for the valuable comments. Added as "WRF v4.3 and its data assimilation system WRFDA v4.3 are used in this study. Two convective cases are studied in the study: 14 June in 2020 (called Case 1; Fig. 1a) and 6 August in 2018 (denoted as Case 2; Fig. 1b). For case 1, the model domain contains 500×471 with a 3 km horizontal grid spacing, and 50 vertical levels. For case 2, the model domain contains 723 × 691 with a 3 km horizontal grid spacing, and 50 vertical levels. The specific applications of physical parametrizations are as follows: the WRF Double-Moment 6-Class Microphysics (WDM6) scheme, the Rapid Radiative Transfer Model (RRTM) long wave radiation scheme (Mlawer et al., 1997), the Dudhia short-wave radiation scheme (Dudhia, 1989), the Yonsei University (YSU) boundary layer scheme (Hong et al., 2006), and the Noah Land Surface Model (Chen and Dudhia, 2001) for land surface process scheme. No cumulus parameterization scheme is employed. As shown in Table 1, three data assimilation (DA) experiments are conducted to evaluate the effects of all retrieval methods in the study. For all three DA experiments, the initial and lateral boundary conditions are provided by the NCEP Global Forecast System (GFS) data. Besides, the specific flowchart is presented in the Fig. 2. The radar observations used in two cases undergo a series of preprocessing and quality control procedure, including anomaly detection, velocity de-aliasing, and so on. The observation errors of radar radial velocity and radar reflectivity are set to 2 m  $\ensuremath{\text{s}}^{-1}$  and 5 dBZ, respectively." in the section 3.

**Minor points**

- 1. It is a little bit confusing to use the "the positive impact is not promising" here.
- Reply: Thanks for the valuable comments. It is revised as "However, due to the absence of ice phase particles, the scheme showed limited effectiveness in deep moist convection cases dominated by cold-cloud processes."
- 2. It is recommended to include topographic information in the simulation domain map shown in Figure 1 to provide additional geographic context.

Reply: Thanks for the valuable comments. Fig.1 has been modified.

Fig. 1. The simulated area of (a) Case 1 and (b) Case 2, with the detecting ranges of the Nanjing radar and Shenyang Radar. Both radars are S-band Doppler radars with a maximum coverage range of 230 km. The radial velocity and reflectivity observations have range resolutions of 250 m and 1000 m, respectively.

 More information about the radar observations used in the data assimilation should be provided, including details such as the type of radar, spatial and temporal resolution, quality control procedures.

Reply: Thanks for the valuable comments.

Added as "Besides, the specific flowchart is presented in the Fig. 2. The radar observations used in two cases undergo a series of preprocessing and quality control

procedure, including anomaly detection, velocity de-aliasing, and so on. The observation errors of radar radial velocity and radar reflectivity are set to  $2 \text{ m s}^{-1}$  and 5 dBZ, respectively." in the section 3.

Fig. 1. The simulated area of (a) Case 1 and (b) Case 2, with the detecting ranges of the Nanjing radar and Shenyang Radar. Both radars are S-band Doppler radars with a maximum coverage range of 230 km. The radial velocity and reflectivity observations have range resolutions of 250 m and 1000 m, respectively.

4. The wind vector arrows in Figure 6 are difficult to discern clearly. Please consider adjusting the arrow color, thickness, or scale to enhance visibility against the background.

Reply: Thanks for the valuable comments. Fig. 6 has been modified.

Fig. 6. The cross sections of relative humidity (shading; units: %), radar reflectivity (black contours starting at 40 dBZ; units: dBZ), and wind vectors for (a) EXP temp, (b) EXP bg and (c) EXP tempbg along the line a1-a2. These are 1-hour forecasts initialized at 1501 UTC.

5. The description of the Equitable Threat Score (ETS) is incomplete. Please provide more details on the calculation method.

Reply: Thanks for the valuable comments.

Added as "In this paper, Equitable Threat Score (ETS) is used to quantitatively evaluate the forecast effect of heavy precipitation in each group of experiments. The specific calculation formula of ETS is as follows:

$$ETS = \frac{A - R}{A + B + C - R},\tag{13}$$

$$R = \frac{(A+C)\times(A+B)}{A+B+C+D},$$
(14)

where A, B, C, and D are the number of hits, the false alarms, the misses, and the correct negatives. The R means the probability to have a correct forecast by chance." in the section 4.1.

6. The discussion of  $\theta$  behavior in convectively unstable environments aligns with theoretical expectations in the Figure 12. While this background is useful, the section could be condensed to focus more sharply on novel aspects of the study.

Reply: Thanks for the valuable comments. It is modified with "Fig. 12 displays the vertical cross sections of the pseudo-equivalent potential temperature ( $\theta$ se), wind components, and reflectivity at 2100 UTC for EXP\_temp, EXP\_bg, and EXP\_temp-bg. All three data assimilation (DA) experiments exhibit a high-low-high vertical distribution of  $\theta$ se. It suggests that the vertical structure of the atmosphere is unstable in this region, with dry conditions prevailing in the upper levels and moist conditions in the lower levels. This type of vertical structure is favorable for the development of severe convective weather events. In the middle layer, there is a zone with relatively high  $\theta$ se value for EXP\_bg and EXP\_temp-bg. Specifically, a warm-core structure is identified near 123.85°N, accompanied by strong upward motion. This results in the release of unstable energy indicate that a severe convective system is continuously developing. Additionally, compared with EXP\_bg, EXP\_temp-bg yields a more extensive and deeper updraft column."